# Electron Density and Its Relation with Electronic and Optical Properties in 2D Mo/W Dichalcogenides

**DOI:** 10.3390/nano10112221

**Published:** 2020-11-08

**Authors:** Pingping Jiang, Marie-Christine Record, Pascal Boulet

**Affiliations:** 1Aix-Marseille University, UFR Sciences, CNRS, MADIREL, 13013 Marseille, France; pingping.jiang@etu.univ-amu.fr (P.J.); pascal.boulet@univ-amu.fr (P.B.); 2Aix-Marseille University, UFR Sciences, University of Toulon, CNRS, IM2NP, 13013 Marseille, France

**Keywords:** two-dimensional materials, DFT calculations, QTAIM, vdW homo- and hetero-structures, structure-properties relationships

## Abstract

Two-dimensional MX_2_ (M = Mo, W; X = S, Se, Te) homo- and heterostructures have attracted extensive attention in electronics and optoelectronics due to their unique structures and properties. In this work, the layer-dependent electronic and optical properties have been studied by varying layer thickness and stacking order. Based on the quantum theory of atoms in molecules, topological analyses on interatomic interactions of layered MX_2_ and WX_2_/MoX_2_, including bond degree (BD), bond length (BL), and bond angle (BA), have been detailed to probe structure-property relationships. Results show that M-X and X-X bonds are strengthened and weakened in layered MX_2_ compared to the counterparts in bulks. X-X and M-Se/Te are weakened at compressive strain while strengthened at tensile strain and are more responsive to the former than the latter. Discordant BD variation of individual parts of WX_2_/MoX_2_ accounts for exclusively distributed electrons and holes, yielding type-II band offsets. X-X BL correlates positively to binding energy (E_b_), while X-X BA correlates negatively to lattice mismatch (lm). The resulting interlayer distance limitation evidences constraint-free lattice of vdW structure. Finally, the connection between microscopic interatomic interaction and macroscopic electromagnetic behavior has been quantified firstly by a cubic equation relating to weighted BD summation and static dielectric constant.

## 1. Introduction

Ever since the discovery of isolated graphene through mechanical exfoliation [1], other graphene-like and two-dimensional (2D) materials, especially transition metal dichalcogenides (TMDs), have attracted significant research interest, in great part due to their unique structure-property relationships. The versatile applications of TMDs have been demonstrated in different fields, such as electronics, optoelectronics, energy storage and photocatalysis [2,3,4,5], despite ongoing challenges in device design and fabrication techniques [6]. Their typical honeycomb-like lattice structures consisting of single or few atomic layers are fundamental for their physical and chemical properties [7]. A wide range of polymorphs and stacking polytypes of TMDs have been defined by the composition, lattice configuration and dimensionality [4]. Among them, the group VI TMDs with formula MX_2_ (M = Mo, W; X = S, Se, Te) stand out thanks to their favorable light-matter interaction, carrier mobility and external radiative efficiency [8,9,10,11,12]. Indeed, the phase stability, electronic structure, and semiconducting property strongly depend on the filling state of M-d orbitals [4,13,14]. As such, the covalent intralayer M-X bonds are the main focus of existing studies [15,16]. However, the interlayer X-X bonds have weak van der Waals (vdW) in nature due to the chemically saturated X atoms, are seldom examined despite being the prime reason for creating atomically thin materials through chemical or mechanical exfoliation [17].

By definition, the layered MX_2_ is divided into three polymorphs, the 1T, 2H, and 3R phases, where letters designate a trigonal, hexagonal, or rhombohedral phase and numbers give the repeating unit of the X-M-X motif [18]. The dissimilar electronic behaviors between phases, particularly at the interface, have driven lots of researches on their layer-dependent properties via experimental and theoretical methods. Mono- and few-layered MX_2_ are strongly distorted, which affects the stability to a certain extent [4,19]. Eda et al. [20] found the immunity of chemically exfoliated MoS_2_ monolayer to structural distortion and local pressure, even though the 1T and 2H phases coexist and form a coherent interface. Additionally, the mechanical strength is linearly dependent on the charge transfer from M to X despite their strong covalent bonding [15]. Huang et al. [16] found that the M-X bond strength plays a key role in phonon propagation, as a result of the competition between ionicity and covalency. Cheng et al. [21] observed a thickness-dependent phase transition and optical behavior of MoS_2_ film when hydrostatic pressure is applied. With layer thinning, the blue shift of the bandgap and photoluminescence (PL) have been detected due to the quantum confinement in a layered d-electron system, presenting a tunable band transition [22,23,24,25]. Moreover, a weak contribution of interlayer interaction to the valence band splitting of bulk WS_2_ was found by Latzke et al. [26].

Sequence stacking two different MX_2_ monolayers by vdW forces in either vertical or lateral directions [27,28] offers supplementary degrees of freedom for heterostructure modulation, thus introducing unexpected properties that are beyond those of individual constitutents [28,29,30]. Heterointerface state has strong implication for predicting physicochemical properties [31]. For instance, the lack of dangling bonds and inactive lone-pair electron at layer-terminated surfaces, and the free of lattice constraint at interfaces help to keep the layer stability against external conditions, irrespective of stacking order [4,32,33]. Previously, substrate, strain, electric field, doping, and defect engineering have been performed on 2D MX_2_ [34,35,36,37]. Su et al. [38] found that stronger bonding between MoS_2_ and substrate results in weaker photoluminescence intensity. Additionally, after subsequent gold nanoparticles deposition, the resulting stress would undermine the Mo-S bonding, thus destabilizing the MoS_2_ film [33].

To date, the majority of studies have focused on the structural, electronic, and optical properties of MX_2_ with or without strain [14,39,40,41], whereas our understanding of the electron density features of 2D homo- and heterostructures with vdW binding forces lags far behind, not to mention the strain- and layout-correlated electrostatic interactions in sublayers. Likewise, the possible relationship between optical properties and electron density features of vdW structures has never been studied, to the best of our knowledge. The role of vdW interactions in ionic and semiconductor solids has been extensively discussed both qualitatively and quantitatively [42,43]. A straightforward supposition is to assume a minor effect of vdW force on the overall cohesive properties in solids [44]. However, the correctness and universality in applying this intuition to 2D and bulk TMDs materials are debatable, since the vdW interaction plays a key role in their band structure evolution [26,45]. Thorough investigations are needed. In the quantum theory of atoms in molecules (QTAIM) [46], the distribution of electron density and the interatomic interaction (defined as bond degree [47]), are correlated to single and group atomic contributions to molecular polarizability. Additionally, the frequency-dependent polarizability per crystal unit cell is a summation of all individual constituents [44,48] and related to the dielectric constant, as expressed in the Clausius-Mossotti relation [49]. Since the macroscopic electromagnetic response and the microscopic chemical bond nature are linked by relative permittivity [50], it is safe to presume that individual bond degree contributions would influence the total dielectric function, as expressed by the dielectric constant as a bulk parameter.

Until now, the common-X [28,30,51] and different-X [40,52,53] heterostructures have been widely studied, but the assembly of two MX_2_ monolayers with different sublayer M and varying X, totaling nine heterostructures, has not been yet considered. Since the full image of interlayer bonds in multi- and hetero-layers and the evolution of electrostatic interactions with composition and dimensionality remain unknown, probing the contribution of both interlayer and intralayer bonds in a wide range of structures is of great interest for further understanding of the structure-property relationships of 2D vdW materials. In this work, three major objectives were addressed by DFT and QTAIM: (i) determination of electronic-dependent structural, optical and bonding properties of MX_2_-based homo- and heterostructures where M = Mo, W and X = S, Se, Te; (ii) detailing and comparing the effect of layer thickness, stacking schemes and lattice mismatch on interatomic interactions, bond lengths and bond angles; and (iii) understanding the lattice mismatch effect on the vdW structures as well as the electron excitations and charge carriers displacements in the materials.

## 2. Computational Details

DFT calculations were carried out by a full-potential linear augmented plane wave method (FP-LAPW) as implemented in the program WIEN2k [54]. To precisely characterize the exchange-correlation energy, the WC-generalized gradient approximation (GGA) functional [55], the modified Becke–Johnson exchange potential (mBJ) functional [56] and the van der Waals corrected optB88-vdW functional [42] were applied and compared. During the lattice optimization and relaxation, the convergence criteria of total energy and force on each atom were set to 10^−5^ Ry and 1 mRy/Bohr, respectively. The R_mt_K_max_ parameter has been set to 7. Additionally, the first Brillouin zone was sampled with a mesh of 1500 ***k***-points using Monkhorst-Pack grids [57]. The lattice constants *a* and *c* and bandgaps of hexagonal P6_3_/*mmc* MoX_2_ and WX_2_ (X = S, Se, Te) were calculated using WC-GGA and optB88-vdW functionals, and compared with other experimental and theoretical data, as summarized in Appendix A. Lattice constants calculated by WC-GGA functional fit better to experimental ones than those calculated by optB88-vdW functional despite the underestimated bandgaps. By using mBJ functional, bandgaps have been improved and agree well with experimental ones due to the accurate semi-local atomic exchange potential [56]. Our parameter settings have been validated and will be used in the following calculations.

The mono, bi-, and trilayered [MX_2_]_n_ slabs in the form of 4×4×1 supercells were built as layer thickness *n* goes from one to three, respectively. In the meantime, the nine WX_2_/MoX_2_ heterostructures were modeled by vertically stacking WX_2_ and MoX_2_ monolayers. Considering boundary conditions and lattice coherence, the in-plane lattice constants of WX_2_/MoX_2_ were set to the average values of those of bulk MoX_2_ and WX_2_ to minimize the effect of lattice mismatch, since their elastic properties in monolayers are similar, such as Young’s modulus in [15,58]. A 20 Å thickness of vacuum has been added atop to separate free surfaces, avoiding interaction between periodic images. The 10×10×1
***k***-meshes were used in the supercell cases. Structure relaxation was performed by using WC-GGA and optB88-vdW functionals until the force on each atom converged to 1 mRy/Bohr. Based on the electron density (*ρ*) obtained from WIEN2k, the topological properties of *ρ* for each modeled structure were investigated with the program CRITIC2 [59]. In the QTAIM, the real space is partitioned into atomic basins, and each basin has only one nucleus which is surrounded by the surface of zero-flux electron density (∆*ρ* = 0). By applying boundary conditions, the local total (*H*), kinetic (*G*), and potential (*V*) energy density can be obtained by integrating over each atomic basin. For more details about the QTAIM method refer to our previous works [60,61].

## 3. 2D Multilayers

### 3.1. Stability, Electronic, and Optical Properties

Figure 1a shows the atomic configurations of mono-, bi- and trilayered [MX_2_]_n_ supercells. Their stabilities are examined by the formation energy $E_{form}$, which is given by:(1)$$E_{form}=E_{slab}-\underset{x=M,X}{\sum }N_{x}E_{x}^{bulk},$$
where $E_{slab}$ is the total energy of the considered slab and $N_{x}$, $E_{x}^{bulk}$ are the number and bulk energy of atoms involved in the slab, respectively. Interlayer distances between metal (M) and chalcogen (X) atoms from the adjacent X-M-X units, i.e., ③ M’-X and ⑥ M″-X’ in Figure 1a, have been compared with the counterparts in bulks. Their differences $\Delta d_{n,bulk}$ and the above yielded $E_{form}$ are gathered in Figure 1b. All layered and bulk MX_2_ are stable at zero Kelvin because of their negative $E_{form}$. In the meantime, the negative $\Delta d_{n,bulk}$ indicates a reinforced binding strength between adjacent layers in [MX_2_]_n_ compared to those in bulks. Since the bilayer and bulk structure have similar unit cells (two layers of X-M-X unit), the gap between their $E_{form}$ would tell the relative stability. Hence, the WTe_2_ bilayer is seen as the least stable, which corresponds to its smallest $\Delta d_{n,bulk}$.

The mutual compromise between bandgap and absorption, which govern voltage and photocurrent, respectively [2], underlines the significance of adjusting bandgap and, hence, absorption startup through layer thickness modification. By adding a vacuum layer atop of layered structures, the uneven interactions of atoms near to and away from the surface will cause physical and chemical deviations from the relevant bulk cases that have intrinsic symmetrical interactions. Thus, apart from the structure stability, further investigations on electronic properties are indispensable. Band structures along the Γ-Σ-M-K-Ʌ-Γ direction in the first Brillouin zone are plotted in Appendix A. Along with the most possible transition directions, interband energy gaps are obtained, as seen in Figure 2a,b. With the growth of thickness, transition energy at K-K’ direction changes slowly whereas those at K-Ʌ and Γ-Ʌ directions decrease rapidly, yielding a direct bandgap for monolayer and indirect ones for bilayers and trilayers. This holds for MoX_2_ and WX_2_ materials, except for the direct bandgap of MoS_2_ bilayer, which is consistent with the results in [22,58,62]. It is found that the layered and bulk WX_2_ have higher and lower bandgap than MoX_2_, respectively. In particular, the Γ-Ʌ transition prevails in bi- and trilayered MoS_2_, WS_2_, and MoTe_2_, while the K-Ʌ transition dominates for the remaining cases, such that the valence band maximum (VBM) and conduction band minimum (CBM) can be determined. The refractive index and absorption coefficient of [MoS_2_]_n_ (as an example) are shown in Figure 3a,b. The photovoltaic (PV) effect grows with the absorber thickness, and that of bulk MoS_2_ is the highest. In the *xx* direction, the starting point of photon-electron transformation is earlier than that in the *zz* direction with both experiencing a red-shift peak at ~2.2 eV and 4.4 eV, respectively. Moreover, the refractive indices in the xx direction are higher than those in the *zz* direction, whereas the absorption coefficients behave oppositely.

### 3.2. Electron Density Analysis

Appendix A displays the distribution of electron density *ρ* and its Laplacian ∇^2^*ρ* of bulk, mono-, bi- and trilayered MoS_2_, which is used as an example for visualization. The labels “b”, “r”, and “c” represent the bond, ring and cage critical points, respectively. Following the trajectory of gradient path between nuclei, the maximum *ρ* shall be reached and the corresponding coordination is called bond critical point (BCP). At BCP, the total (*H*_BCP_), kinetic (*G*_BCP_), and potential (*V*_BCP_) energy density are functionals of electron density (*ρ*_BCP_) with the relationship of *H*_BCP_ = *G*_BCP_+*V*_BCP_. The magnitude of interatomic interaction is represented by absolute bond degree (BD = *H*_BCP_/*ρ*_BCP_ [47,63]). The distance between nuclei n_1_ and n_2_ in the form of “n_1_-b-n_2_” is named as bond length. Through the BDs vs. |*V*_BCP_|/*G*_BCP_ ratios and BLs of M-X (①, ②, and ⑤ in Figure 1a) and X-X (④ and ⑦ in Figure 1a) bonds in bulk, mono-, bi-, and trilayered MX_2_, as shown in Figure 4, the exact bonding nature to composition and dimensionality can be determined. The signs of ∇^2^*ρ*_BCP_ provide details analogous to those found about *ρ*_BCP_ and have been used to differentiate the closed-shell (CS) and shared-shell (SS) interactions by using the adimensional |*V*_BCP_|/*G*_BCP_ ratio according to the local virial theorem [46] (*h*^2^/16π^2^*m*∇^2^*ρ*_BCP_=2*G*_BCP_+*V*_BCP_, where *h* and *m* are the Planck constant and electron mass, respectively). In addition to the pure CS (|*V*_BCP_|/*G*_BCP_ < 1) and SS (|*V*_BCP_|/*G*_BCP_ > 2) interactions, another region called *transit* CS (1 < |*V*_BCP_|/*G*_BCP_ < 2) has been defined [47]. As shown in Figure 4a, the X-X bond lies in the pure CS region, presenting a local charge-depletion interaction, in other words, forming vdW-like bonding between X atoms. In contrast, the M-X bond lies in the *transit* CS and pure SS regions, presenting the local charge-concentration interactions, namely, forming covalent bonding between M and X atoms.

The bonding features with the growth of layer thickness can be specified from Figure 4a,b. One can notice that, the vdW bond is sensitive to the layer dimensionality and that, as already evidenced in [63] for other kinds of materials, BDs and |V|/G ratios of X-X bonds are linearly correlated. For a given X, BDs of X-X bonds in bulks (marked as stars) are the highest compared with those in bilayers (marked as triangles) and trilayers (marked as circles). The bulk MoX_2_ has higher BD and BL of X-X bond than WX_2_, whereas the bi- and trilayered MoX_2_ have lower BDs and BLs of X-X bonds than WX_2_. This coincides with the bandgap results in Figure 3. Meanwhile, as *n* goes from two to three, BDs and BLs of X-X bonds in [MoX_2_]_n_ are barely changed, while those in [WX_2_]_n_ decrease and increase respectively, with a declining speed as X goes from S to Te. Additionally, the bulk and layered MoX_2_ have smaller absolute BDs and BLs of M-X bonds than WX_2_, leading to a stronger W-X bond compared to Mo-X one at any given X, which is consistent with [16]. As *n* goes from one to three, no significant changes are found for the BDs and BLs of M-X bonds. Thus, except for the Mo-Te bond, the rest M-X bonds in [MX_2_]_n_ are slightly stronger but shorter than those in bulks. As X goes from S to Te, the absolute BDs of X-X and M-X bonds decrease despite the discordances in BLs. Given the linear BD responses of X-X bonds to composition and dimensionality, a dominant position of the interlayer vdW bond in layer-dependent properties is foreseen. Based on Gatti’s assumption [64], an atomic expectation value may be equated to a sum of “bond” contributions. In the quest for the structure-properties relationships, we tried to find a relation between the bond degree summation and the dielectric constant. Since in the above discussion structures are fully relaxed and without external perturbation, we have considered the dielectric constant under zero-incident energy ε_1_(0) along the thickening direction. In each unit cell, the numbers of X-X are different, i.e., zero for monolayer, one for bilayer, and two for trilayer. Taking this into account, the weight coefficients *l* and *m* have been used to express the absolute BD summation, which is written as *l*|BD|_M-X_+*m*|BD|_X-X_. Fitting the *l*|BD|_M-X_+*m*|BD|_X-X_ to the ε_1_(0) via equation $y=A+Bx+\cdots +Nx^{n}+O\left(x^{n+1}\right)$, the maximum coefficient of determination R^2^ is achievable by adjusting the *l* and *m* at each equation order. As plotted in Figure 5a, the maximum values of R^2^ for bulks, bilayers, and trilayers can be located as the equation order goes from first to second and to third. Among them, a cubic equation $y=A+Bx+Cx^{2}+Dx^{3}$ could describe the relationship between bond degree and dielectric constant accurately with the maximum R^2^ over 0.99 at *l*/*m* = 0.5 and the lowest Akaike and Bayesian information criteria (AIC and BIC) compared with the linear and quadratic equations, as listed in Table 1. This *l*/*m* ratio indicates that the X-X bond shares more responsibility in dielectric function than the M-X one, and that share keeps constant no matter the layer thickness. In particular, the R^2^ of monolayers are 0.685, 0.974 and 0.999 at the first, second and third orders respectively without adjusting the *l*/*m*, since there is only M-X bond, i.e., *m* = 0. At each layer thickness, the absolute BD summation and the ε_1_(0) are inversely related. An advantage in electron excitation is predictable for the entirely weak interatomic interaction, such that explains robust optical response when incident energy is applied. As layer thickness goes up, the fitting curves shift upwards and rightwards, leading to gradually amplified BD summation and ε_1_(0), which evidences the layer-dependent properties of vdW materials. Despite the BD summation of bulk material is in between those of bilayer and trilayer, the ε_1_(0) of the former is the highest, which is consonant with the refraction and absorption results in Figure 3.

## 4. 2D Heterostructure

### 4.1. Stability, Electronic, and Optical Properties

The WX_2_/MoX_2_ (X = S, Se, Te) heterostructures were assembled with the top WX_2_ and bottom MoX_2_ monolayers taken from their bilayers and denominated from N° 1 to N° 9, as shown in Figure 6a,b. At the interface, X atom from the WX_2_ side is directly placed atop M atom from the MoX_2_ side, since this is the most favorable stacking order as explained in [30,51]. After relaxation, the binding energy (*E*_b_) is given by:(2)$$E_{b}=1/2A\left[E_{WX_{2}/MoX_{2}}-1/2\left(E_{WX_{2}}+E_{MoX_{2}}\right)\right]$$
where $E_{WX_{2}/MoX_{2}}$, $E_{WX_{2}}$, and $E_{MoX_{2}}$ are the energy of heterostructure and its constitutive top and bottom bilayers, respectively, and *A* is the interface area. The in-plane lattice mismatch (Δ*a*/*a*_0_) and *E*_b_ of heterostructures are plotted in Figure 6c,d. As can be seen, most of the heterostructures have vdW-like binding energy under both WC-GGA and optB88-vdW functionals, which are in the range of 19–45 meV/Å^2^. Two energy values are out of that range, namely the most negative *E*_b_ of N°5 (WSe_2_/MoSe_2_) with WC-GGA functional and the least negative *E*_b_ of N°9 (WTe_2_/MoTe_2_) with both two functionals. The more negative *E*_b_, the more stable the structure. The N°5 and N°9 are the most stable and unstable respectively despite their smallest lattice mismatches. Additionally, no dependence has been found between structural stability and lattice mismatch since a larger Δ*a*/*a*_0_ is unnecessarily referring to a smaller *E*_b_ and vice versa, such as the N°2 and N°9 cases.

The band structures and density of states (DOS) of the nine heterostructures are shown in Figure 7. The N°1-2 and N°4-6 have direct bandgaps even though their parent bilayers have indirect ones, as shown in Appendix A. Additionally, the N° 8–9 have indirect bandgaps and the N° 3 and 7 have zero bandgaps. By hetero-stacking, the indirect to direct bandgaps transition becomes possible, as found in [30,52]. The VBMs of N° 1, 5, and 9 are composed of M-d, X-p orbitals of MX_2_ (with M being Mo and W), while the CBMs are composed of Mo-d and X-p orbitals of MoX_2_. Additionally, the VBMs and CBMs of N° 2, N° 3, and N° 6 are composed of W-d, X-p orbitals of WX_2_ and Mo-d, X-p orbitals of MoX_2_, respectively. In contrast, the VBMs and CBMs of N° 4 and N° 7 are composed of Mo-d, X-p orbitals of MoX_2_ and W-d, and X-p orbitals of WX_2_, respectively. A special case is observed for N° 8 whose VBM and CBM are both composed of Mo-d and Te-p orbitals. The MX_2_ monolayers contribute exclusively to either CBM or VBM, forming a built-in separation of electrons and holes. This physical detachment favors the collection but recombination of charge carriers.

At heterointerfaces, the band alignments, including valence (VBO) and conduction (CBO) band offsets, are obtained by formulae:(3)$$\Delta E_{VBO}^{WX_{2}/Mo{X^{\prime }}_{2}}=\Delta E_{VBM,C}^{WX_{2}}-\Delta E_{VBM,C^{\prime }}^{Mo{X^{\prime }}_{2}}+\Delta E_{C,C^{\prime }}$$
and:(4)$$\Delta E_{CBO}=\Delta E_{g}+\Delta E_{VBO}$$
where $\Delta E_{VBM,C}^{WX_{2}}$ and $\Delta E_{VBM,C^{\prime }}^{Mo{X^{\prime }}_{2}}$ are the energy gaps between core levels, i.e., X/X’-1s, and VBMs of bulk WX_2_ and MoX’_2_, respectively, and $\Delta E_{C,C^{\prime }}$ is the binding energy difference between X-1s and X’-1s in WX_2_/MoX_2_ [61,65]. As plotted in Figure 8b, except for N° 8, the WX_2_/MoX_2_ belong to type II (“cliff-like”) band offsets with opposite signs of VBOs and CBOs. This matches the results in [66] that the common-X hetero-system has type-II band alignment. As a result of that, the VBMs and CBMs are contributed by different sides originating from the metal *d* and chalcogen *p* repulsion. For example, the positive VBOs and negative CBOs of N° 1–3, 5–6, and 9 lead to the WX_2_-composed VBMs and MoX_2_-composed CBMs. Upon excitation, the excited electrons will jump from WX_2_ sides to MoX_2_ ones, forming the flow path ❷ in Figure 8a. In contrast, the negative VBOs and positive CBOs of N° 4 and 7 will result in an opposite flow path from MoX_2_ sides to WX_2_ ones, i.e., path ❶ in Figure 8a. An exception case is found for the N° 8 whose VBM and CBM are both contributed by MoTe_2_. This type-I (“spike-like”) band offset will drive the electrons flow within the MoTe_2_ monolayer, forming the path ❸ in Figure 8a.

From the aforementioned observations, the contributions from both the top and bottom monolayers are responsible for its entire performances in electronic and optical fields. As shown in Appendix A, the absorption coefficients of WX_2_/MoX_2_ are over ~10^5^ cm^−1^. To further characterize the PV ability of WX_2_/MoX_2_ thin-film, the spectroscopic limited maximum efficiency (SLME) method [67] has been employed. The input power condition is chosen as the AM 1.5G illumination in the wavelength range of 280–1200 nm. Based on the out-of-plane absorption coefficient, the maximum conversion efficiency *η*_max_ concerning film thickness can be obtained by the output short-circuit current (*J*_SC_) and open-circuit voltage (*V*_OC_). The fast increasing *J*_SC_ and slow decreasing *V*_OC_, originating from charge carrier enrichment and recombination, respectively, give rise to a continuously growing *η*_max_ with the film thickness. As shown in Figure 8c, the *η*_max_ of WX_2_/MoX_2_ all converge to certain values which are over 25%, and accordingly, that with small atomic number is lower than that with large atomic number.

### 4.2. Electron Density Analysis

The distributions of electron density and its Laplacian of WS_2_/MoS_2_ are shown in Appendix A, which is used as an example. The in-plane lattice mismatch (*lm*) in heterostructures and the corresponding signs in the constitutive bilayers are listed in Table 2, where the “+” sign means tensile strain and the “-“ sign means compressive strain. The BDs with respect to |*V*_BCP_|/*G*_BCP_ ratios and BLs of X-X and M-X bonds in heterostructures and the constitutive top and bottom bilayers are plotted in Figure 9 and Figure 10, respectively. Based on the |*V*_BCP_|/*G*_BCP_ ratio, X-X bonds have vdW bonding and M-X bonds have covalent bonding, which is the same as those in primitive layered and bulk structures. Upon the construction of heterostructures, the linear correlation between BDs and |*V*_BCP_|/*G*_BCP_ ratios of X-X bonds is maintained, underlying a linear responsivity of vdW bonding force to *lm*. Irrespective of the X-X bond, its BL (Figure 9d) and *E*_b_ (Figure 6d) are correlated; smaller BL corresponds to less negative *E*_b_. For example, the WTe_2_/MoTe_2_, with the shortest Te-Te bond at the interface, has the least negative *E*_b_ despite its small *lm* since stability is irrelevant to *lm*, as discussed before.

By comparing the bonding nature in heterostructure and its constitutive bilayers, the influence of hetero-stacking on opposite sublayers can be drawn. Hereafter, bonds in the constitutive top and bottom bilayers are marked by signs (‘) and (“), respectively. In top WX_2_ bilayers, the BDs and BLs of S-S (N° 1′, 4′, 7′), Se-Se (N° 2′, 5′, 8′), Te-Te (N° 3′, 6′, 9′) and W-S (N°1′, 4′, 7′) bonds increase as the *lm* goes from “-” to “+”. This holds for S-S (N° 2′’, 3′’), Se-Se (N° 4′’, 5′’, 6′’), Te-Te (N° 7′’, 8′’, 9′’) and Mo-S (N° 1”, 2′’, 3′’) bonds in bottom MoX_2_ bilayers as well. In contrast, the BDs and BLs of W-Se (N°2′, 5′, 8′), W-Te (N° 3′, 6′, 9′), Mo-Se (N° 4′’, 6′’), and Mo-Te (N° 7′’, 8′’, 9′’) bonds decrease and increase, respectively, as the *lm* goes from “-” to “+”. Two exceptional cases are found for the N° 1′’ S-S and N° 5′’ Mo-Se bonds, which corresponds to the abruptly intensified N° 1′’ Mo-S and weakened N° 5′’ Se-Se. The BDs and BLs are positively correlated for X-X and M-S bonds, while negatively correlated for M-Se and M-Te bonds. Compared with X-X bonds, W-X and Mo-X ones have less evident BL changes. When *lm* < 0 and *lm* > 0, the absolute BDs of X-X and M-X bonds go the opposite way, such as N° 3′, 6′, 9′ bonds at WTe_2_ bilayers and N° 1′’, 2′’, 3′’ bonds at MoS_2_ bilayers. Consequently, under tensile strain, X-X and M-S bonds tend to weaken and elongate while M-Se/Te ones tend to strengthen and elongate; under compressive strain, X-X and M-S bonds tend to strengthen and shorten while M-Se/Te ones tend to weaken and shorten. Additionally, as chalcogen atomic number goes up, BDs of M-X bonds first decrease and then increase with continuously increasing BLs, whereas BDs and BLs of X-X bonds decrease continuously. Additionally, the BD is more susceptible to compressive strain than tensile strain, as a result of the expanding BD difference under the former (e.g., N° 7′’ and 8′’ of MoTe_2_) while the shrinking one under the latter (e.g., N° 2′’ and 3′’ of MoS_2_). The discordant BD evolution to strains explains the separate generation of electrons and holes in the opposite monolayers, which will be discussed below.

Given the strain-induced bonding behaviors of the top and bottom bilayers, as illustrated above, the electronic performances of heterostructures can be identified. The N° 1′-3′, 5′-6′, and 9′ X-X bonds in WX_2_ bilayers (Figure 9b) have smaller BDs than the N° 1′’-3′’, 5′’-6′’, and 9′’ ones in MoX_2_ bilayers (Figure 9c), leading to a stronger capability of electron excitation in WX_2_ monolayers than MoX_2_ ones after stacking them together. In the meantime, the absolute BDs of W-X bonds in WX_2_/MoX_2_ (Figure 10a) are lower than those in WX_2_ bilayers (Figure 10b), whereas those of Mo-X bonds in WX_2_/MoX_2_ (Figure 10d) overlap with those in MoX_2_ bilayers (Figure 10e). Therefore, after stacking two monolayers together, the interatomic interaction of W-X bond is undermined while that of Mo-X bond is unchanged. Coupling with the prevalent excitation in X-X bond, a superior electron excitation shall be located in the WX_2_ monolayer, followed by the electron transition to the MoX_2_ monolayer through M-d and X-p orbitals. This is the underlying mechanism for path ❷ mentioned above. Similarly, the smaller BDs of N° 4′’ and 7′’ X-X bonds than those of N° 4′ and 7′ ones would end up with a superior capability in electron excitation at MoX_2_ parts. Additionally, the absolute BDs of Mo-X bonds decrease while those of W-X bonds are barely changed after stacking. For the same reason, abundant electrons could be located at the MoX_2_ monolayer and flow to the WX_2_ monolayer. Nevertheless, the smaller BD of N° 8′’ X-X bond than that of N° 8′ one brings the preference of electron excitation to the MoX_2_ monolayer. Though the BD of W-X bond increases further than that of Mo-X one after stacking, the preeminent position of X-X bond in electron excitation will drive the electron flowing within the MoX_2_ monolayer. Thus, path ❶ and ❸ are explained.

From Figure 9d, the BLs of X-X bonds in heterostructures are close to the average values of those in the constitutive bilayers, and within the range of 4.45–4.70 Å. Additionally, no consistency is found between BDs and BLs of X-X bonds as the *lm* varies. For instance, in the N° 1–3 cases where MoS_2_ is fixed as the bottom monolayer, the BDs of X-X bonds decline linearly while the BLs first increase and then decrease with the *lm*. To further elucidate the relationship between bond geometry and binding energy at interfaces, the bond angles (BA) ∠n_1_bn_2_ of X-X and M-X bonds in the form of “n_1_-b-n_2_” are discussed, as plotted in Figure 11. As can be seen, the BAs of X-X bonds in the constitutive bilayers are close to 180° and overlap with each other, whereas those in heterostructures are lower than 180° and interestingly, are inversely correlated to the *lm*. On the contrary, the BAs of W-X and Mo-X bonds have barely visible differences between those in heterostructures and two relevant bilayers. The X-X bond is keen on lattice mismatch by the BCP migration towards the atomic end with high electronegativity. In other words, the “b” in the form above shifts to either n_1_ or n_2_. The X-X bond encounters lattice strain by BA adjustment, such that the interlayer distance is maintained within a certain range. Thereupon, the mechanisms for vdW heterostructure free of lattice constraint are concretized by the positively correlated bond length and binding energy, and the negatively-correlated bond angle and lattice mismatch.

To examine the applicability of the above cubic relationship between absolute BD summation *l*|BD|_M-X_+*m*|BD|_X-X_ and static dielectric constant *ε*_1_(0) in WX_2_/MoX_2_ heterostructures and the constitutive bilayers with interfacial strains, the polynomial equations with different orders are also used by changing the *l*/*m* ratio, as plotted in Figure 12a. It needs mentioning that in WX_2_/MoX_2_, the contribution of M-X bonds to the BD summation is set to the average value of Mo-X and W-X bond, namely, *l*|BD|_M-X_ = *l*/2(|BD|_Mo-X_+|BD|_W-X_). As equation order goes from first to third, the R^2^ increases accompanied by an enlarging *l*/*m* ratio. At the corresponding maximum value, the AIC and BIC of the third equation order are the lowest compared with those of other orders, as listed in Table 1. The fitting curve and coefficients are shown in Figure 12b and Table 1, respectively, on behalf of the well-described cubic relationship (R^2^ = 0.960). Compared with the *l*/*m* of primitive bulk and layered MX_2_ (*l*/*m* = 0.5), that of structures subjected to strain (*l*/*m* = 0.3) is smaller, as well as the R^2^, which could be explained by the strain-induced deformation in interatomic interactions of X-X bonds as well as the uneven interactions between W-X and Mo-X bonds. Once strain exists, a superior capability of X-X bond in responding to the electromagnetic field is foreseen by sharing more responsibility for dielectric function than M-X bond. Its prompt response is coherent with the linear reaction to strain and hetero-stacking, as shown in Figure 9a. The decrease of interatomic interaction goes along with the increase of dielectric constant. This is the reason for heterostructure with a higher chalcogen atomic number having better optical behavior, as the SLME in Figure 8c illustrates. Indeed, quantifying the relationship between optical and bonding properties is of great meaning for interpreting theunderlying bonding mechanism and predicting the macroscopic development of vdW material whose lattice and strain changes. Accordingly, the resulting correlation of structural engineering to interatomic interaction allows us to anticipate the potentiality to achieve equivalent optical response but with fewer material expenses.

## 5. Conclusions

DFT calculations of MX_2_ homo- and heterostructures, the layer-dependent structural, electronic and optical properties have been performed. In layered MX_2_, a broadening bandgap and its indirect-direct transition happen with the thinning of thickness. The widespread bandgaps as X goes from S to Te envisages designs of lightweight and diverse PV cells. Based on the QTAIM, electron density analyses are detailed to connect microscopic interaction with macroscopic electromagnetic behaviors. The relationship between optical and bonding properties can be formulated by a cubic equation relating to weighted BD summation and static dielectric constant. Irrespective of structure and strain, X-X bonds lie in charge-depletion region with linearly correlated BDs and |*V*|/*G* ratios, whereas M-X bonds lie in charge-accumulation region with barely changed BDs as layer thickness varies. In effect, the layer-dependent electronic and optical properties are mainly attributed to the X-X bond due to its prompt responsivity to strain and hetero-stacking. The WX_2_/MoX_2_ have type-II band offsets with the maximum conversion efficiency of over 25% owing to the exclusive distribution of electrons and holes in separate sublayers. Positively correlated vdW bond length and binding force, and negatively-correlated vdW bond angle and lattice mismatch are the reasons for vdW structures free of lattice constraint, maintaining interlayer distances within ranges. Our findings contribute to increasing the knowledge of the TMDs materials and offering a new approach for the development of the 2D stacked thin-film solar cells.

## Figures and Tables

**Figure 1 nanomaterials-10-02221-f001:**
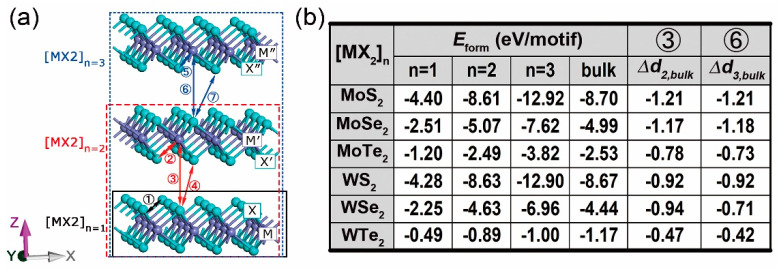
Scheme of [MX_2_]_n_ supercells with M = Mo, W and X = S, Se, Te when *n* equals to 1, 2, and 3 (**a**). Metal and chalcogen atoms in the first, second and third X-M-X units are marked as [M, X], [M’, X’], and [M″, X″], respectively. Formation energy $E_{form}$ of bulk and [MX_2_]_n_ slabs and differences of M-X distances in slabs and bulks, $\Delta d_{n,bulk}$ (**b**).

**Figure 2 nanomaterials-10-02221-f002:**
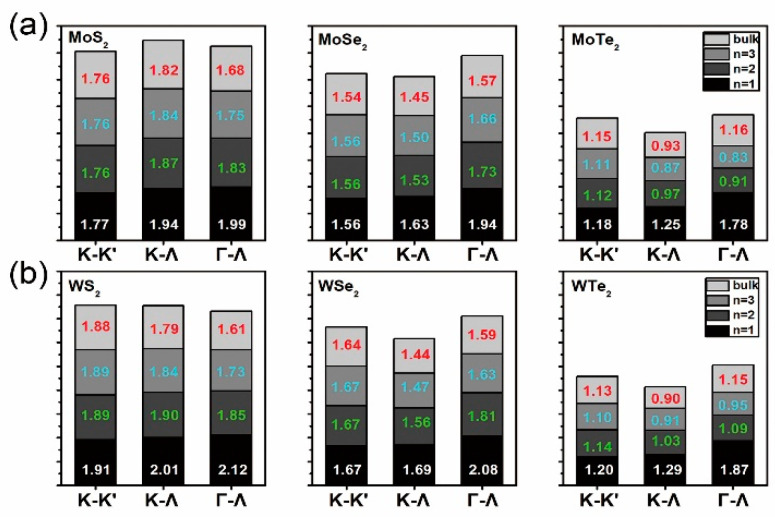
Direct (K-K’) and indirect (K-Ʌ and Γ-Ʌ) band transition energy in eV of bulk (red texts), mono- (white texts), bi- (green texts), and trilayered (cyan texts) MoX_2_ (**a**) and WX_2_ (**b**).

**Figure 3 nanomaterials-10-02221-f003:**
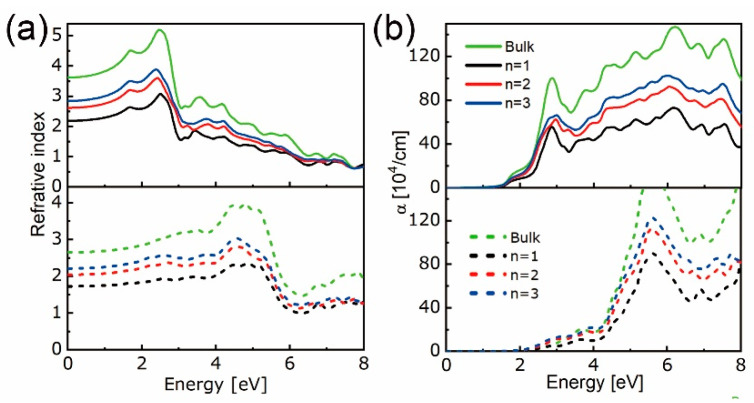
Refractive index (**a**) and absorption coefficient α (**b**) of bulk and layered [MoS_2_]_n_ in the in-plane (top panel) and out-of-plane (bottom panel) directions.

**Figure 4 nanomaterials-10-02221-f004:**
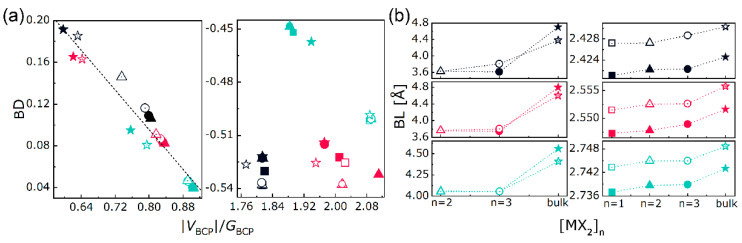
Bond degrees (BD) vs. |*V*_BCP_|/*G*_BCP_ ratios (**a**) and bond lengths (BL) (**b**) of X-X (left panel) and M-X (right panel) bonds in [MX_2_]_n_. Symbols used in plots, solid: MoX_2_; hollow: WX_2_; black, red, and teal colors: X = S, Se and Te; squares, triangles, circles, and stars: *n* = 1, 2, 3, and bulk.

**Figure 5 nanomaterials-10-02221-f005:**
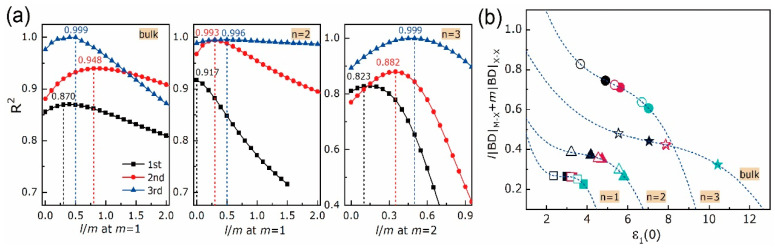
Results of *l*|BD|_M-X_+*m*|BD|_X-X_ vs. ε_1_(0) of bulk and [MX_2_]_n_: (**a**) fitting coefficient of determination R^2^ vs. *l*/*m* ratio as equation order goes from first to second and to third; (**b**) fitting curves of cubic equation $y=A+Bx+Cx^{2}+Dx^{3}$ at maximal R^2^. Symbols used in (**b**), solid: MoX_2_; hollow: WX_2_; black, red and teal colors: X = S, Se and Te; squares, triangles, circles and stars: *n* = 1, 2, 3, and bulk.

**Figure 6 nanomaterials-10-02221-f006:**
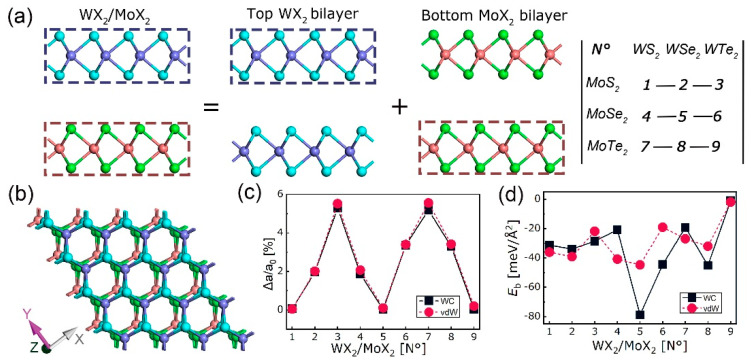
Schemes of WX2/MoX_2_ (X = S, Se, Te) heterostructures: (**a**) Side view; (**b**) top view; (**c**) in-plane lattice mismatch Δ*a*/*a*_0_ in % and (**d**) binding energy *E*_b_ in meV/Å^2^ at heterointerfaces.

**Figure 7 nanomaterials-10-02221-f007:**
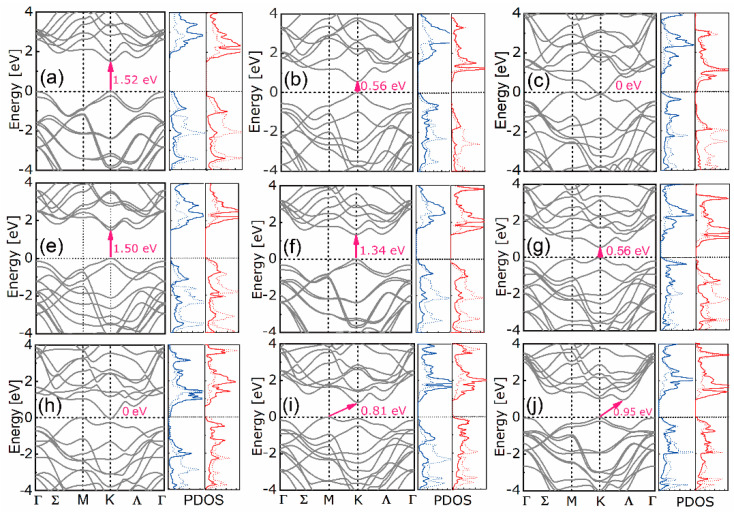
Band structures and density of states (in WX_2_, W-d: solid blue; X-p: dash blue; in MoX_2_, Mo-d: solid red; X-p: dash red) of N° 1–9 WX_2_/MoX_2_ (**a**–**j**).

**Figure 8 nanomaterials-10-02221-f008:**
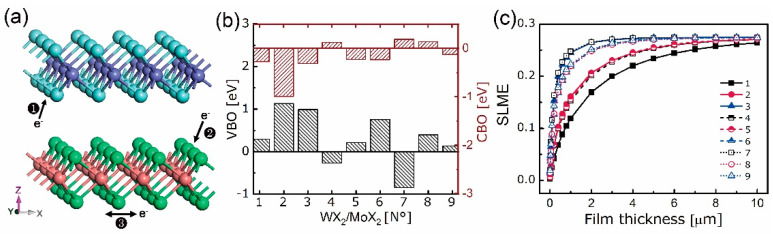
Electron flow path (**a**), valence (black) and conduction (bordeaux) band offsets (**b**) and spectroscopic limited maximum efficiency (**c**) of N° 1–9 WX_2_/MoX_2_. Path ❶, ❷, and ❸ represent the electron flow from MoX_2_ to WX_2_, WX_2_ to MoX_2_, and within MoTe_2_, respectively.

**Figure 9 nanomaterials-10-02221-f009:**
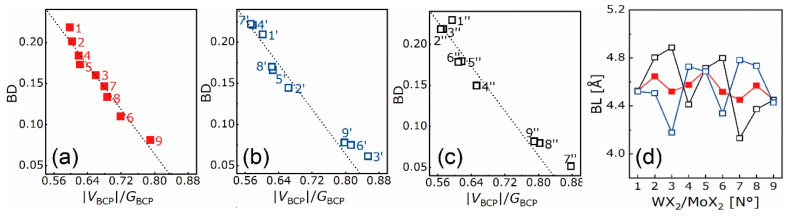
BDs vs. |*V*_BCP_|/*G*_BCP_ ratios of X-X bonds in N° 1–9 WX_2_/MoX_2_ (red) (**a**), top WX_2_ bilayers (blue) (**b**), bottom MoX_2_ bilayers (black) (**c**), and their BLs (**d**).

**Figure 10 nanomaterials-10-02221-f010:**
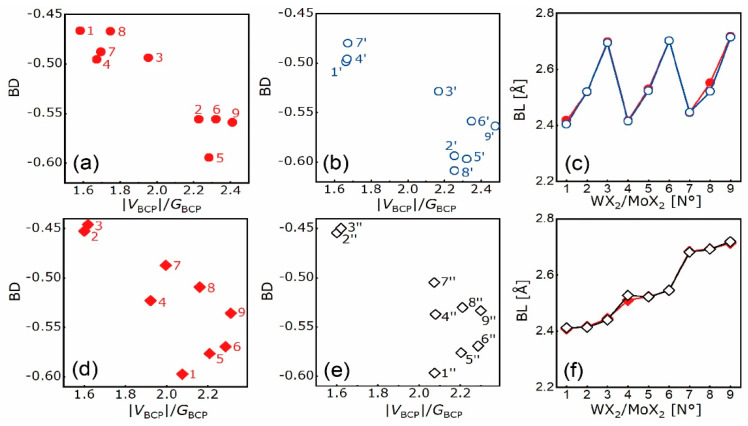
BDs vs. |*V*_BCP_|/*G*_BCP_ ratios of W-X bonds in N°1-9 WX_2_/MoX_2_ (red) (**a**), top WX_2_ bilayers (blue) (**b**) and their BLs (**c**). BDs vs. |*V*_BCP_|/*G*_BCP_ ratios of Mo-X bonds in WX_2_/MoX_2_ (red) (**d**), bottom MoX_2_ bilayers (black) (**e**), and their BLs (**f**).

**Figure 11 nanomaterials-10-02221-f011:**
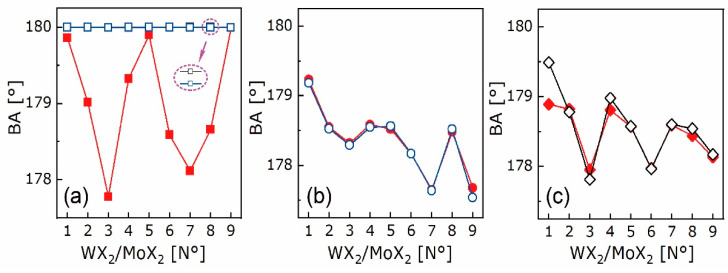
Bond angles (BA) of X-X (**a**), W-X (**b**), Mo-X (**c**) bonds in N° 1–9 WX_2_/MoX_2_ (red), top WX_2_ bilayers (blue), and bottom MoX_2_ bilayers (black).

**Figure 12 nanomaterials-10-02221-f012:**
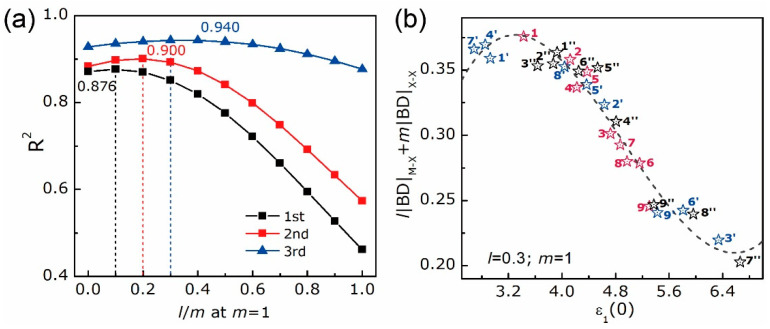
Results of *l*|BD|_M-X_+*m*|BD|_X-X_ vs. ε_1_(0) of WX_2_/MoX_2_ and the constitutive bilayers: (**a**) fitting coefficient of determination R^2^ vs. *l*/*m* ratio as equation order goes from first to second and to third; (**b**) cubic fitting curves at maximum R^2^ (=0.940) with red, blue, and black colors representing heterostructures, top WX_2_ (‘) and bottom MoX_2_ (“) bilayers, respectively.

**Table 1 nanomaterials-10-02221-t001:** Polynomial fitting of *l*|BD|_M-X_+*m*|BD|_X-X_ vs. ε_1_(0) via equation $y=A+Bx+\cdots +Nx^{n}+O\left(x^{n+1}\right)$ of bulk, layered [MX_2_]_n_ and WX_2_/MoX_2_ heterostructures. Fit quality evaluation criteria, including R^2^, AIC and BIC, as well as coefficients A, B, C, D at the first, second and third equation orders.

	Equation Order	R^2^	AIC	BIC	A	B	C	D
[MX_2_]_n=1_	1st	0.685	−36.76	−37.17	−2.59 × 10^−2^	3.39 × 10^−1^		
	2nd	0.974	−49.68	−50.3	3.26 × 10^−2^	1.78 × 10^−1^	−3.32 × 10^−2^	
	3rd	0.999	−67.74	−68.58	1.07 × 10^0^	−8.54 × 10^−1^	3.04 × 10^−1^	−3.61 × 10^−2^
[MX_2_]_n=2_	1st	0.917	−29.4	−29.82	−4.66 × 10^−2^	5.56 × 10^−1^		
	2nd	0.993	−44.34	−44.96	1.63 × 10^−1^	1.32 × 10^−1^	−1.96 × 10^−1^	
	3rd	0.996	−56.68	−57.51	8.75 × 10^−1^	−3.69 × 10^−1^	9.45 × 10^−2^	−8.46 × 10^−3^
[MX_2_]_n=3_	1st	0.823	−21.89	−22.31	−5.90 × 10^-2^	4.71 × 10^−1^		
	2nd	0.882	−22.84	−23.46	7.91 × 10^−2^	9.25 × 10^−2^	−1.40 × 10^−2^	
	3rd	0.999	−60.44	−61.27	1.85 × 10^0^	−5.47 × 10^−1^	9.55 × 10^−2^	−6.10 × 10^−3^
Bulk	1st	0.870	−27.53	−27.94	−2.76 × 10^−2^	3.81 × 10^−1^		
	2nd	0.948	−27.36	−27.98	7.98 × 10^−1^	−9.49 × 10^−3^	−1.66 × 10^−3^	
	3rd	0.999	−54.23	−55.06	1.09 × 10^0^	−2.30 × 10^−1^	2.89 × 10^−2^	−1.34 × 10^−3^
WX_2_/MoX_2_	1st	0.876	−102.3	−100.3	−5.46 × 10^−2^	5.63 × 10^−1^		
	2nd	0.900	−104.9	−101.9	3.87 × 10^−1^	2.40 × 10^−2^	−8.45 × 10^−3^	
	3rd	0.940	−114.7	−110.7	−5.82 × 10^−1^	6.98 × 10^−1^	−1.60 × 10^−1^	3.03 × 10^−3^

**Table 2 nanomaterials-10-02221-t002:** In-plane lattice mismatch (*lm*) in % in WX_2_/MoX_2_ and the signs in the constitutive top WX_2_ (‘) and bottom MoX_2_ (‘’) bilayers. The “-” and “+” represent compressive and tensile strain, respectively.

WX_2_/MoX_2_	N°	1	2	3	4	5	6	7	8	9
	*lm*	0.06	1.96	5.30	1.88	0.02	3.36	5.19	3.30	0.04
Top [WX_2_]_n=2_	X	S	Se	Te	S	Se	Te	S	Se	Te
	N°	1′	2′	3′	4′	5′	6′	7′	8′	9′
	*Sign*	-	-	-	+	-	-	+	+	-
Bottom [MoX_2_]_n=2_	X	S	S	S	Se	Se	Se	Te	Te	Te
	N°	1′’	2′’	3′’	4′’	5′’	6′’	7′’	8′’	9′’
	*sign*	+	+	+	-	+	+	-	-	+

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
