# Peer review of "Electron Density and Its Relation with Electronic and Optical Properties in 2D Mo/W Dichalcogenides"

_nanomaterials, 2020, doi:10.3390/nano10112221_

Round 1

Reviewer 1 Report

Jiang et al. report a DFT calculation of several layer-dependent electronic and optical properties of TMDs. Despite the topic treated in the manuscript has scientific interest, the main message and the potential applicability of the work are not clear to this reviewer. Thus, there are several concerns preventing my recommendation for publication of the current manuscript in Nanomaterials:

  1. The topological view is one of the main claims of the manuscript, and it even starts the title. However, I have not found any topological parameter (e.g. topological invariant) all along the paper. I find this issue particularly relevant.
  2. The authors show several regression fittings along the paper (e.g., Figures 4 and 5). No insights are provided into the physical interpretation of the parameters involved in the resulting functions (A, B, C, etc.). Particularly, the authors claim in the conclusions section that “topological analyses are detailed to connect microscopic interaction with macroscopic electromagnetic behaviors [line 437]”. However, this connection reduces to a cubic law. A detailed exploration of the implications of this fact are missing to this referee. This point is related to the previous one (1).
  3. The figures need to be considerably improved, as they are not easy to inspect, extremely small, and some labels are even missing. The text reads disconnected. For instance, Figure 3b is introduced before Figures 2b, 2c and 3a. I feel that Section 2 could be entirely moved to the Supplementary Materials. The number of references might be excessive.

So overall, I feel the content should be substantially improved and arranged in a more organized way, stating clearly the main claims/messages of the work.

Author Response

Jiang et al. report a DFT calculation of several layer-dependent electronic and optical properties of TMDs. Despite the topic treated in the manuscript has scientific interest, the main message and the potential applicability of the work are not clear to this reviewer.

Authors’ answer: Our work contributes to the understanding of the optical responses in the 2D Mo/W dichalcogenides heterostructures. In particular, by coupling DFT calculations and electron density topological analysis, we evidenced the flow paths of the electrons. This kind of investigation has never been done so far and can serve as a basis for the design of materials with improved performance. This information has been added in the revised manuscript.

Thus, there are several concerns preventing my recommendation for publication of the current manuscript in Nanomaterials:

Reviewer’s question (RQ): The topological view is one of the main claims of the manuscript, and it even starts the title. However, I have not found any topological parameter (e.g. topological invariant) all along the paper. I find this issue particularly relevant.

Authors’ answer (AA): The topology we are referring to is that of the electron density as described in Bader’s “quantum theory of atoms in molecules” (QTAIM). In his theory, the topological investigation relies on the finding of peculiar points in the electron density and their characterization, namely the bond critical points that are saddle points, the ring critical points, the cage critical points and the attractors that are atoms. Except at the attractors, the gradient of the electron density vanishes at the critical points. It is also possible to univocally partition the space around atoms in the so-called atomic basins into which the density can be integrated to obtain data of chemical relevance (e.g. the atoms charge). The critical points have specific signature (ω,λ), where ω is the rank of the electron density Laplacian and λ is the sum of the Laplacian eigenvalue signs. Among these points, the bond critical point is of particular interest as they characterize the chemical bonding nature between two bonded atoms as we report in our paper.  In this respect, the critical points are kind of invariants of the electron density, and once identified, using the Critic2 package, we use the properties of the electron density Laplacian to characterize the nature of the interatomic bonds.

To avoid misunderstanding or unmet expectations in this respect, we have changed the title to “Electron density and its relation with optical properties in 2D Mo/W dichalcogenides” and the sections 3.2 and 4.2 named “topological property of the electron density” have been renamed as “Electron density analysis”. The keyword “topological property” has been replaced by “QTAIM”. When ambiguous, the term “topological” has been replaced by a more explicit one in the revised version.

RQ: The authors show several regression fittings along the paper (e.g., Figures 4 and 5). No insights are provided into the physical interpretation of the parameters involved in the resulting functions (A, B, C, etc.). Particularly, the authors claim in the conclusions section that “topological analyses are detailed to connect microscopic interaction with macroscopic electromagnetic behaviors [line 437]”. However, this connection reduces to a cubic law. A detailed exploration of the implications of this fact is missing to this referee. This point is related to the previous one (1).

AA: Since in a previous work (H. Yang et al., Comput. Theor. Chem., 2020, 1178, 112784) bearing on other kinds of materials a linearity between BDs and |V|/G ratios was found, we fitted the data presented in Figure 4 to test the validity of this relation. Regarding the fit in figure 5, it is based on the Gatti assumption that tells that an atomic expectation value may be equated to a sum of “bond” contributions. In the quest for the structure-properties relationships, we tried to find a relation between the bond degree summation and the dielectric constant. We have added these information in the revised paper.

RQ: The figures need to be considerably improved, as they are not easy to inspect, extremely small, and some labels are even missing. The text reads disconnected. For instance, Figure 3b is introduced before Figures 2b, 2c and 3a. I feel that Section 2 could be entirely moved to the Supplementary Materials. The number of references might be excessive.

AA: The figures have been improved and the sequence of figures reference in the text has been reordered properly. Section 2 corresponds to the description of the methods used with optimized lattice constants and calculated band gaps. In the revised manuscript we have shortened this section by transferring the table with the calculated results to the supplementary materials.

So overall, I feel the content should be substantially improved and arranged in a more organized way, stating clearly the main claims/messages of the work.

AA: The section titles have been changed for more clarity. We have deleted the heading words introducing the paragraphs of the introduction. The objectives of the work were defined in the original manuscript but admittedly were blurred by the density of the text. We have improved their visibility in the revised version by bringing them at the end of the introduction. We also have better put forward the originality of the results.

Reviewer 2 Report

Overall the paper is very hard to read because of the complicated language used.

Figures have a lot of information (almost too much), but the font and markings are small and not very clear (e.g. Fig. 2 (a))

The paper needs to be structured better (clear sections / subsections / formulas separately etc...). At current form it is almost impossible to digest.

Author Response

Overall the paper is very hard to read because of the complicated language used.

Authors’ answer (AA): we have worked on the writing of the paper to make clearer the main objectives, results and applicability of our work.

Reviewer’s question (RQ) : Figures have a lot of information (almost too much), but the font and markings are small and not very clear (e.g. Fig. 2 (a))

AA: The figures have been improved for better readability. Also, we have divided Figure 9 into four parts for better understanding.

RQ : The paper needs to be structured better (clear sections / subsections / formulas separately etc...). At current form it is almost impossible to digest.

AA: We have deleted the heading words introducing the paragraphs of the introduction and for more clarity the section and subsection titles have been changed. In addition, for improving the readability of the manuscript we have separated the equations from the flow of the text.

Reviewer 3 Report

JIANG et al., Topological view of structure-property relationships 3 on 2D Mo/W dichalcogenides

Here authors investigate, two-dimensional MX2 homo- and heterostructures for electronics and optoelectronics applications. In this work, the layer-dependent electronic and optical properties have been studied by varying layer thickness and stacking order. They show that bonds are strengthened and weakened in layered compared to the counterparts in bulks. It also depends on the compressive strain or tensile strain and also affects the band offsets.

The calculations were performed in a very detailed manner and showed very good results. Therefore, the manuscript should be considered after the suggested revisions.

  1. In the calculation for the refractive index and absorption coefficient in MX2, what is the effect on the presence of anisotropy and strain in the system?
  2. In the considered WX2/MoX2 heterostructures, what is the effect of moiré potential due to hetero-bilayer and twist angle of homo-layer on the calculated properties? How the calculated binding energy Eb is affected?
  3. type-II band offsets with the maximum conversion efficiency of over 25% are calculated. It should be compared with experimental results and discussed the gap between the experiments and theory.

Author Response

Here authors investigate, two-dimensional MX2 homo- and heterostructures for electronics and optoelectronics applications. In this work, the layer-dependent electronic and optical properties have been studied by varying layer thickness and stacking order. They show that bonds are strengthened and weakened in layered compared to the counterparts in bulks. It also depends on the compressive strain or tensile strain and also affects the band offsets.

The calculations were performed in a very detailed manner and showed very good results. Therefore, the manuscript should be considered after the suggested revisions.

Reviewer’s question (RQ): In the calculation for the refractive index and absorption coefficient in MX2, what is the effect on the presence of anisotropy and strain in the system?

Authors’ answer (AA): According to our calculation results, as the incident photon energy increases, the refractive index and absorption coefficient in the in-plane direction are higher than those in the out-of-plane direction. With the growth of out-of-plane thickness, the refraction and absorption increase and are at a decreasing rate. We are planning to study the effect of strain on the optical properties of MX2 slabs in our next step.

RQ: In the considered WX2/MoX2 heterostructures, what is the effect of moiré potential due to hetero-bilayer and twist angle of homo-layer on the calculated properties? How the calculated binding energy Eb is affected?

AA: In our work, the supercells are modelled with no twist angle. We understand the importance of stacking schemes on mechanical, electronic and optical properties of heterostructures, such as the exciton diffusion length in Ref.[Choi et al. Moiré Potential Impedes Interlayer Exciton Diffusion in van der Waals Heterostructures]. We are very interested in continuing to work on this in the future.

RQ: type-II band offsets with the maximum conversion efficiency of over 25% are calculated. It should be compared with experimental results and discussed the gap between the experiments and theory.

We value the reiewer’s suggestion very much. We would like to proceed with the comparison of state-of-art and experimental conversion efficiency of WX2/MoX2 more explicitly in our next step.

Reviewer 4 Report

I read the manuscript entitled “Topological view of structure-property relationships on 2D Mo/W dichalcogenides” by P. Jiang et al, submitted for a publication in nano materials. The manuscript reports structural, electronic and optical properties of MX2 (M= Mo,W) chalcogenides in homo- and heterostructures, calculated using DFT. The authors have discussed in details how the structural properties at the local scale including the bond lengths and layers thickness etc are related with the physical properties of these systems. I think the work is a useful addition to the already available large amount of literature in the specialized field of potentially important layered chalcogenides. I recommend publication of this manuscript. I do have some minor technical questions:

1) Calculated bandgap underestimated and I am wondering if this has any effect on the conclusions of the present work regarding optical properties.

2) Presumably the calculations are performed at 0 K and I am wondering if there will be any significant change to be expected at room temperature.

3) What is the energy cutoff in the calculation, perhaps I missed it.

4) I think the quality of some of the figures is poor and notations are too small to be readable (see, particularly e.g. fig.7,8).

Author Response

I read the manuscript entitled “Topological view of structure-property relationships on 2D Mo/W dichalcogenides” by P. Jiang et al, submitted for a publication in nano materials. The manuscript reports structural, electronic and optical properties of MX2 (M= Mo,W) chalcogenides in homo- and heterostructures, calculated using DFT. The authors have discussed in details how the structural properties at the local scale including the bond lengths and layers thickness etc are related with the physical properties of these systems. I think the work is a useful addition to the already available large amount of literature in the specialized field of potentially important layered chalcogenides. I recommend publication of this manuscript. I do have some minor technical questions:

Reviewer’s question (RQ): Calculated bandgap underestimated and I am wondering if this has any effect on the conclusions of the present work regarding optical properties.

Authors’ answer (AA): although the bandgaps are generally underestimated, as we are making comparisons between alike materials, the conclusions should not be too much affected.

RQ: Presumably the calculations are performed at 0 K and I am wondering if there will be any significant change to be expected at room temperature.

AA: As the reviewer says, the calculations are done at 0 K, but we have no means to estimate the changes that may occur due to finite temperature. Nonetheless, some authors recently developed a model of the dielectric functions of the Au, Ag and Cu metals based on the scattering of electrons by electrons, phonons, lattice defects and impurities that accounts for temperature [M. Xu et al., PRB, 96, 115154 (2017)]. Their results show that, the effect of temperature is rather small on the dielectric response. It is however difficult to transpose their results on simple metals to complex semiconductors such as ours.

RQ: What is the energy cutoff in the calculation, perhaps I missed it.

AA: In the FP-LAPW formalism the planewave cutoff energy is replaced by the RmtKmax parameter that has been set to 7. This information has been added in the “computational details” section.

RQ: I think the quality of some of the figures is poor and notations are too small to be readable (see, particularly e.g. fig.7,8).

AA: the quality of the figures has been improved and the font sizes increased.

Reviewer 5 Report

This is a computational work on MoX2 structures using density functional theory (DFT) and quantum theory of atoms in molecules (QTAIM). The authors using the well-known DFT code Wien2K to calculate electronic properties such as band structure and densities of states and the CRITIC2 program to calculate QTAIM properties, such as electron density and Laplacian at critical points. This is a very detailed work and highly scientifically sound. The paper is easy for the reader to follow. It should be accepted with the following minor revision:

The authors are recommended to perform a  few single energy calculations with a hybrid DFT functional and compare with their results from GGA calculations.

Author Response

This is a computational work on MoX2 structures using density functional theory (DFT) and quantum theory of atoms in molecules (QTAIM). The authors using the well-known DFT code Wien2K to calculate electronic properties such as band structure and densities of states and the CRITIC2 program to calculate QTAIM properties, such as electron density and Laplacian at critical points. This is a very detailed work and highly scientifically sound. The paper is easy for the reader to follow. It should be accepted with the following minor revision:

The authors are recommended to perform a  few single energy calculations with a hybrid DFT functional and compare with their results from GGA calculations.

Authors’ answer: We appreciate the reviewer’s recommendation. This will be our future work to deal with the bandgap underestimation problems with GGA functionals.

Round 2

Reviewer 1 Report

The authors have improved their manuscript and resolved several issues from the previous set of reports. However, there are still some concerns that remain unaddressed and that prevent my recommendation for publication of the current manuscript in Nanomaterials:

According to the authors, “the relationship between optical and bonding properties can be formulated by a cubic equation relating to weighted BD summation and static dielectric constant”. This claim is based on the R-squared value of the fitting. “In particular, the R2 of monolayers are 0.606, 0.956 and 0.998 at the first, second and third orders respectively”. Yet, R-squared by itself does not indicate if a regression model provides an adequate fit to a set of data. An adequate model can have a low R2 value while a biased or overfitted model can have a higher R2 value. The fitting parameters shown in the inset in Figure 5b show that D is close to zero in all the cases, so I would say that a quadratic law, which is simpler and therefore easier to interpret, would still adjust fairly well. For this kind of situations, there are criteria for model selection (LASSO regression, BIC, AIC, and others). Hence, what kind of regression have the authors performed and how have they chosen the cubic law? I feel this issue is too little elaborated for the importance it has on this work.

In this regard, I asked about the interpretation of the parameters in the cubic law (namely A, B, C, D) in my previous report. This interpretation is still missing in the current version of the manuscript. In relation to this question, the authors have pointed in their response that in a previous work (H. Yang et al., Comput. Theor. Chem., 2020, 1178, 112784) a linearity between BDs and |V|/G ratios was found. In that reference (by the way with some co-authors in common with the present manuscript submitted to Nanomaterials), an interpretation of the slope coefficient is given. I quote verbatim: “the slope |a1| reflects resistance to deformation of the electronic shells (reciprocal of polarizability) under pressure and can be considered as the rigidity of the interaction between atoms”. What do A, B, C, D represent in the present case?

Anisotropy is an important optical feature of 2D Mo/W dichalcogenides. Too little is said in the calculation results of the optical properties about the effect of anisotropy in the system.

The authors have modified the title to “Electron density and its relation with optical properties in 2D Mo/W dichalcogenides”. I would say the properties explored are not only optical, but also electronic, etc. Consequently, I would suggest a rather broader adjective, such as “physical properties”.

In general, I think the quality of some of the figures is still poor. In particular, I find figure labels, markings and/or insets small and not very clear (Fig. 6d, Fig. 7 and others). Mainly, the paper is hard to follow and is not well structured yet.

Author Response

The authors have improved their manuscript and resolved several issues from the previous set of reports. However, there are still some concerns that remain unaddressed and that prevent my recommendation for publication of the current manuscript in Nanomaterials:

According to the authors, “the relationship between optical and bonding properties can be formulated by a cubic equation relating to weighted BD summation and static dielectric constant”. This claim is based on the R-squared value of the fitting. “In particular, the R2 of monolayers are 0.606, 0.956 and 0.998 at the first, second and third orders respectively”. Yet, R-squared by itself does not indicate if a regression model provides an adequate fit to a set of data. An adequate model can have a low R2 value while a biased or overfitted model can have a higher R2 value. The fitting parameters shown in the inset in Figure 5b show that D is close to zero in all the cases, so I would say that a quadratic law, which is simpler and therefore easier to interpret, would still adjust fairly well. For this kind of situations, there are criteria for model selection (LASSO regression, BIC, AIC, and others). Hence, what kind of regression have the authors performed and how have they chosen the cubic law? I feel this issue is too little elaborated for the importance it has on this work.

Author’s Answer: As illustrated in our article, there is a presumption that the summation of interatomic interaction (BD) is correlated with electromagnetic behavior, which is chosen as the static dielectric constant ε(0). Therefore, we try to use the polynomial equation fitting to validate our presumption. The “R-square” has been chosen as an indicator to look into the dependence between BD summation and ε(0). The parameter D is not zero, we have adjusted the decimal place and used scientific notation. As the reviewer suggested, we have added the fit quality evaluation criteria AIC and BIC into the new Table 1. As we can see, the AIC and BIC of the cubic fitting are the lowest compared with those of the linear and quadratic fittings. Together with the high “R-square”, we have a good reason to say that the cubic law is well-founded.

In this regard, I asked about the interpretation of the parameters in the cubic law (namely A, B, C, D) in my previous report. This interpretation is still missing in the current version of the manuscript. In relation to this question, the authors have pointed in their response that in a previous work (H. Yang et al., Comput. Theor. Chem., 2020, 1178, 112784) a linearity between BDs and |V|/G ratios was found. In that reference (by the way with some co-authors in common with the present manuscript submitted to Nanomaterials), an interpretation of the slope coefficient is given. I quote verbatim: “the slope |a1| reflects resistance to deformation of the electronic shells (reciprocal of polarizability) under pressure and can be considered as the rigidity of the interaction between atoms”. What do A, B, C, D represent in the present case?

Author’s Answer: The possible physical meanings of A, B, C, D are not clear yet. This is the reason for using polynomial fitting. We will still work on it.

Anisotropy is an important optical feature of 2D Mo/W dichalcogenides. Too little is said in the calculation results of the optical properties about the effect of anisotropy in the system.

Author’s Answer: In this work, the optical anisotropy of 2D Mo/WX2 has been studied by calculating the refractive indexes, absorption coefficients and loss tangent in two perpendicular directions.

The authors have modified the title to “Electron density and its relation with optical properties in 2D Mo/W dichalcogenides”. I would say the properties explored are not only optical, but also electronic, etc. Consequently, I would suggest a rather broader adjective, such as “physical properties”.

Author’s Answer: We would like to change the title to “Electron density and its relation with electronic and optical properties”.

In general, I think the quality of some of the figures is still poor. In particular, I find figure labels, markings and/or insets small and not very clear (Fig. 6d, Fig. 7 and others). Mainly, the paper is hard to follow and is not well structured yet.

Author’s Answer: The figure qualities have been improved. The titles of section 4.1, 4.2 have been changed to “stability, electronic and optical properties”, and the figure sequence.

Reviewer 2 Report

I'm happy with the modifications made by authors.

Author Response

Author's answer: We are thankful to the reviewer for his careful analysis of our paper.

Round 3

Reviewer 1 Report

The results are not clearly presented yet, the quality of the figures is still poor and some of my (and other referee's) previous comments remain unadressed.